# Grounding Visual Illusions in Language:
# Do Vision-Language Models Perceive Illusions Like Humans?

**Yichi Zhang**
University of Michigan
zhangyic@umich.edu

**Jiayi Pan**[*]
University of California, Berkeley
jiayipan@berkeley.edu

**Yuchen Zhou**[*]
The New School
zhoua926@newschool.edu

**Rui Pan**
Northwestern University
rui.pan@kellogg.northwestern.edu

**Joyce Chai**
University of Michigan
chaijy@umich.edu

## Abstract

Vision-Language Models (VLMs) are trained on vast amounts of data captured by humans emulating our understanding of the world. However, known as visual illusions, human's perception of reality isn't always faithful to the physical world. This raises a key question: do VLMs have the similar kind of illusions as humans do, or do they faithfully learn to represent reality? To investigate this question, we build a dataset containing five types of visual illusions and formulate four tasks to examine visual illusions in state-of-the-art VLMs. Our findings have shown that although the overall alignment is low, larger models are closer to human perception and more susceptible to visual illusions. Our dataset and initial findings will promote a better understanding of visual illusions in humans and machines and provide a stepping stone for future computational models that can better align humans and machines in perceiving and communicating about the shared visual world. The code and data are available at github.com/vl-illusion/dataset.

## 1 Introduction

It's well established that human perceptual systems are susceptible to visual illusions, which are defined as "consistent and persistent discrepancies between a physical state of affairs and its representation in consciousness" (Day, 1984). Figure 1 shows a well-known example - the checker shadow illusion (Adelson, 1995). Here, a cylinder on the checker board creates a shadow on the board. Human viewers are directed to look at the two squares A and B as shown in Figure 1(a). To most normal-sighted people, they will perceive that square A is

---

*Work done while the author was an undergraduate research assistant at the University of Michigan.

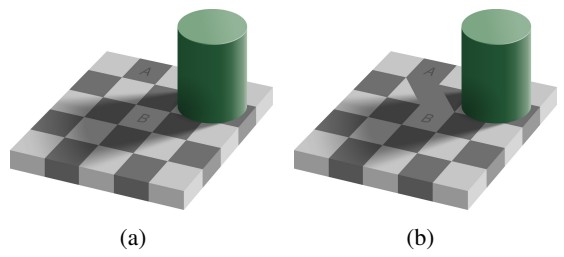

Figure 1: The checker shadow illusion (Adelson, 1995).

darker than square B. However, the reality is, the color pixels of A and B are exactly the same, as shown in Figure 1(b). This example demonstrates that while the physical attributes of A and B are the same, from humans' eyes, they may look different, which may further influence how language is used to describe these objects.

Motivated by human visual illusion phenomena, recent years have seen an increasing amount of work in machine visual illusions (Gomez-Villa et al., 2019, 2020; Hirsch and Tal, 2020; Sun and Dekel, 2021; Lonnqvist et al., 2021). These previous works were solely based on vision, for example, evaluating how the internal representation from a computer vision model can be used as a proxy of stimulus compared to human's stimulus shift under illusions. Most previous experiments were conducted in a case-by-case manner, without addressing general behaviors through a systematic investigation.

Different from these previous works, this paper studies visual illusion from a new angle, in the context of language communication. Language comprehension and language production are tightly linked to how we perceive the visual world. Back to Figure 1(a), when two people are observing the figure together, due to their likely shared illusion, the

expression "the darker square" will lead to the same reference of square A. But when a human communicates with a machine, will the machine also interpret "the darker square" as square A? Given the rise of large vision-language models (VLM), it's important to understand whether these VLM models have a similar tendency to visual illusions, and to what degree they may align with human vision. The answers to these questions will further impact the alignment of the grounded language communication between humans and machines.

To address these questions, we created a new visual illusion dataset covering five different categories from the cognitive literature. Based on the dataset, we created a benchmark, Grounding Visual Illusion in Language (GVIL), which consists of four subtasks: Same-Difference Question Answering (*SameDiffQA*), Referential Question Answering (*RefQA*), Attribute Question Answering (*AttrQA*), and Referential Localization (*RefLoc*) to assess machines' alignment with the human under visual illusions. We specifically evaluated four state-of-the-art vision-language models: Unified-IO (Lu et al., 2022), OFA (Wang et al., 2022), LLaVA (Liu et al., 2023) and InstructBLIP (Dai et al., 2023). Our results have shown that these four models mostly do not align with human vision illusions, especially for QA-based tasks. However, for the *RefLoc* task, these models (especially ones with larger parameters) have demonstrated an impressive alignment with humans.

To our knowledge, this is the first work that takes language into consideration to study machine visual illusion. There are two main contributions of this work. First, this investigation provides an initial understanding of the alignment between human and machine visual illusions. Such understanding will enable techniques for a better communicative grounding in situated language communication and help develop more reliable embodied agents in the future. Second, unlike using internal representations to explain illusions, language can be used as a proxy to demonstrate whether and how machine illusions match or mis-match with the human illusion. This benchmark will not only facilitate computational models for better human agent alignment, but also provide tools for scientific understanding of visual illusion in both humans and machines.

## 2   Related Work

**Human Visual Illusion**   Visual illusions in humans are instances where human subjective perceived properties, such as color or size, deviates from their true physical characteristics (Carbon, 2014). This underscores the fact that the human brain doesn't perfectly replicate physical features; rather, it integrates contextual information and prior knowledge to form the perceptual experiences (Carbon, 2014).

Visual illusions can affect human behavior in multiple ways. Research has shown that human action cannot resist visual illusions (Gentilucci et al., 1996; Franz, 2001; Carey, 2001), so is language comprehension and language production. Such findings catalyze inquiries regarding the capability of models to comprehend language instructions based on human perceptions and align them with human intent.

**Machine Visual Illusion.**   Previous studies have significantly advanced our ability to examine visual illusions by providing systematic data and tools. These efforts include the introduction of tools for calculating and generating illusory images systematically (Hirsch and Tal, 2020; Fan and Zeng, 2023), the development of open-source software with a parametric framework for controlled illusion generation (Makowski et al., 2021), and the proposal of a framework synthesizing new visual illusions using automatic differentiation techniques (Gomez-Villa et al., 2022). With the goal of evaluating machine visual illusions, prior research (Gomez-Villa et al., 2019, 2020; Afifi and Brown, 2019; Benjamin et al., 2019) has also demonstrated that convolutional neural networks trained on ImageNet or low-level vision tasks can be misled by certain visual illusions, similar to human responses. These works have formed a foundation for scalable and reproducible research on machine illusions.

Unlike prior research focusing exclusively on vision models, our study introduces a novel and unique angle by presenting the first dataset offering natural language annotations for the evaluation of machine-processed visual illusions. This work intends to bridge the current gap and facilitate future evaluations of vision-language models concerning their alignment with human visual illusions. This novel perspective illuminates future improvements in human-machine alignment and promotes the crucial role of human language as the interaction inter-

face with machines.

**Foundation Vision-Language Models.** Recent advancements in foundational vision-language models (VLMs) have shown impressive results across a broad spectrum of tasks (OpenAI, 2023; Wang et al., 2022; Lu et al., 2022; Alayrac et al., 2022; Radford et al., 2021). These models, acting as user interfaces, interact with users through both language and visuals, necessitating a deep understanding of human intent and an alignment with human values to make them more useful. While previous research has primarily focused on language-based uni-modal alignment problems (Ouyang et al., 2022; Kosinski, 2023), our work offers a fresh perspective. Centered on the intersection of VLM's perception capability and human cognitive biases, we investigate to what degree they can understand humans and align with human intentions under visual illusions.

## 3 The Grounding Visual Illusion in Language (GVIL) Benchmark

To facilitate our investigation, we created a benchmark for evaluating machine visual illusions in the context of grounded communication. This benchmark is built upon a set of images with visual illusions. Each image consists of two objects which may look different to humans but are actually identical in their pixels. This setup has two advantages. First, the definition of illusion is clear and non-ambiguous, thus it is easy to measure whether the machine has a similar illusion as humans. Secondly, the multi-object setup naturally supports the evaluation of language grounding, such as evaluating whether the machine can select the object an expression grounds to under the illusion (i.e., square A is what "the darker square" grounds to in Figure 1(a)).

### 3.1 Data Collection

Our dataset encapsulates five distinct types of illusions, each reflecting different elements of human physiological and cognitive processes (Gregory, 1997; Kitaoka, 2010; Robinson, 2013). Table 1 displays a sample of each illusion type, along with a detailed description.

These illusions can be categorized into two broad areas: color and geometric illusions. For color illusions, we adopt the classifications of color constancy, assimilation, and simultaneous contrast (MacEvoy, 2005). In terms of geometric illusions,

| Color Constancy | |
|---|---|
| 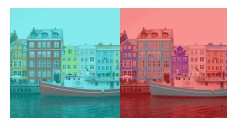 | The red ship on the left still looks red after applying a blue filter, the blue ship on the right still looks blue after applying a red filter, even though the RGB colors of both ships are the same. |
| **Color Assimilation** | |
| 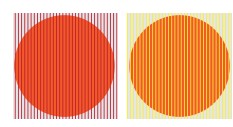 | The two circles have the same color, while the one on the left looks red (due to its neighbor/foreground) and the one on the right looks orange. |
| **Color Contrast** | |
| 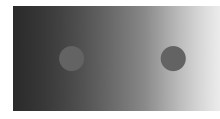 | The two grey circles have the same color, while the one on the left looks lighter and the one on the right looks darker. |
| **Geometrical Relativity** | |
| 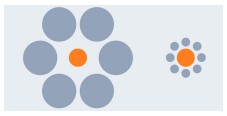 | The two orange circles have the same size, while the one on the left looks smaller and the one on the right looks bigger. |
| **Geometrical Perspective** | |
| 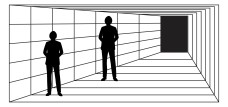 | The two people have the same height, while the one on the left looks shorter and the one on the right looks taller. |

Table 1: Example illusion from each category and the corresponding explanations.

we only included distortions among the four categories in Robinson's illusion classification in order to fix the need for a comparative assessment. The illusions we used to generate our data include Delboeuf (Delboeuf, 1865), Ebbinghaus, and Jastrow illusions (Jastrow, 1892) for relativity, and Müller-Lyer (Müller-Lyer, 1889) and Ponzo illusions (Ponzo, 1911) for perspective distortion. The following explanations give an overview of the human perception phenomena underlying each category:

- **Color Constancy** refers to phenomenon where the color of an object remains constant perceptually, despite changes in lighting conditions.

- **Color Assimilation** shows how two adjacent color patches influence each other's perceptual appearance, reducing the perceived color difference.

- **Color Contrast.** The perceived color of an object shifts opposite to the color of its surroundings

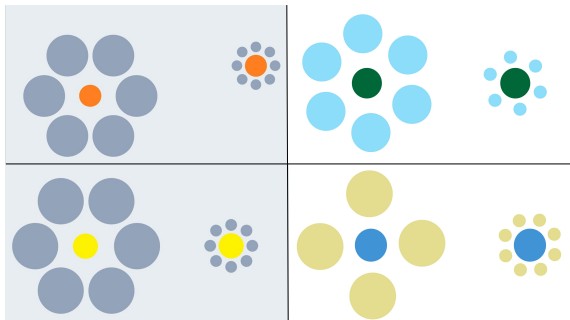

Figure 2: Data augmentation examples for the Ebbinghaus illusion.

- **Geometrical Relativity** refers to the distortion in the perceived shape or size of an object due to the influence of surrounding oobjects.
- **Geometrical Perspective** reflects the tendency of our visual system to perceive perceptually distant objects as larger than nearby ones of the same size.

For each illusion type, we first collected several root images from the literature (Todorović, 2020) and online resources[1]. We manually identify attributes that can be changed without impacting the effect of illusion (e.g., the color of objects in geometric illusions, or the position of objects in color illusions), and edit them to create more illusion instances of the same type, to enrich the number of images in our dataset. We show some augmentation examples in Figure 2.

The statistics of our dataset is shown in Table 2. Note that since this dataset is only used for the evaluation purpose, i.e., to assess machine's alignment with human in visual illusion, we chose quality over quantity. The dataset is modest in size as each instance is carefully selected (or augmented) based on cognitive literature. Nonetheless, our infrastructure is also designed to foster the continuous development and augmentation of this dataset, allowing for community contributions, for instance. It will become an important resource to not only support a better understanding of machine/human visual illusion, but also facilitate the adaptation of computational models to visual illusions.

### 3.2 Benchmark Tasks

We define four vision-language tasks targeting different model capabilities.

**Same-Different Question Answering (*SameDiffQA*)** aims at evaluating the ability of recognizing

[1] https://michaelbach.de/ot/

| Category | #Root | #Image | #Instance |
|----------|-------|--------|-----------|
| Color Constancy | 3 | 6 | 96 |
| Color Assimilation | 5 | 34 | 544 |
| Color Contrast | 3 | 30 | 480 |
| Geometrical Relativity | 3 | 20 | 320 |
| Geometrical Perspective | 2 | 10 | 160 |
| Total | 16 | 100 | 1600 |

Table 2: Dataset statistics.

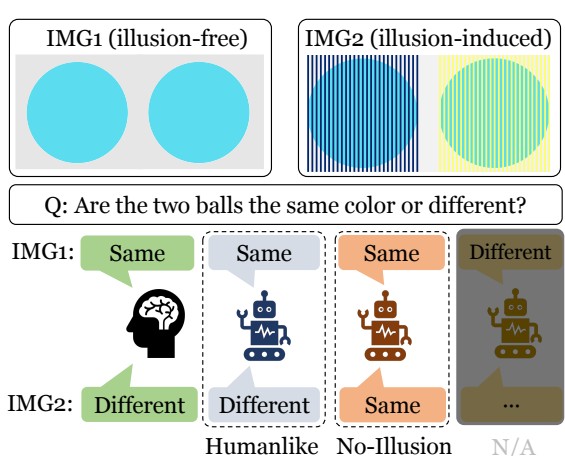

Figure 3: Illustration of the *SameDiffQA* setup. For each instance, the model is asked about its perception of an object property across two images, one illusion-free and one illusion-induced. For valid illusion evaluation, the model must initially identify identical properties in the illusion-free image.

illusions. As shown in Figure 3, each question concerns a pair of images (IMG1 and IMG2). One image (IMG1) is illusion-free where two objects (blue balls) are identical in color. The other image (IMG2) is induced with an effect of illusion where two balls appear in different colors (blue on the left and green on the right) although their pixels are the same as in IMG1. The model is tasked to answer whether two objects are the same color for each of the images. From a human's perspective, the answer would be "Same" to IMG1 and "Different" to IMG2 due to the visual illusion. If the model gives the answer 'Same" to IMG1 and "Different" to IMG2, then the answers align with human's answers and therefore the model is considered "human-like". If the model gives "Same" to both images, it implies that the model is faithful to reality and does not perceive the same illusion as humans do. If the model answers "Different" to IMG1, it means it lacks basic ability to correctly perceive reality and these cases are considered not applicable to our illusion evaluation.

While *SameDiffQA* focuses on the detection of the presence of illusions, we design three tasks to examine how well do machines align with humans when communication happens under the influence of illusions. Since it is reported that models tend to take shortcut by giving an answer purely based on the text question without looking at the image (Goyal et al., 2017), we propose a paired test to reduce the evaluation bias. As shown in Figure 4, each instance comes with two images: one original illusion image (IMG1) and one image IMG2 that flips the objects from the original image (IMG1) in a way that will also invert the answer to the question.

Specifically, we evaluate the following three aspects:

**Referential Question Answering (*RefQA*)** tests the human-likeness of referring to objects under the illusion. In the question, the object of interest is referred to by a property affected by the illusion, and the machine is asked to select the object from two options, e.g., select either left or right for the ball that looks blue, in IMG1 and IMG2.

**Attribute Question Answering (*AttrQA*)** tests the human-likeness to describe the attribute of objects under the illusion. The question is about describing a visual attribute (e.g. color) influenced by the illusion of a selected object, and we provide two answer options to select from.

**Referential Localization (*RefLoc*)** tests the human-likeness of localizing the referred object under the illusion. Given a referential expression that makes sense to humans under the effect of illusion (but may not be faithful to reality), the model needs to predict a bounding box for the object the expression is referring to. For each referential query, we consider the machine's response to be humanlike only when the pair of responses from the two images both match with those from human's. This enforces that a humanlike response from machines has to be grounded in the illusion image.

To create this benchmark, we annotate the collected illusion images with natural language questions and the corresponding answers that humans will give under illusions. To support the study of language grounding, we also annotate the referring expressions for each of the objects with the corresponding bounding box, where the expressions are formed under illusions. We provide several paraphrases for all the language annotations to help the evaluation be more robust to models that are sensitive to language forms. All the images and

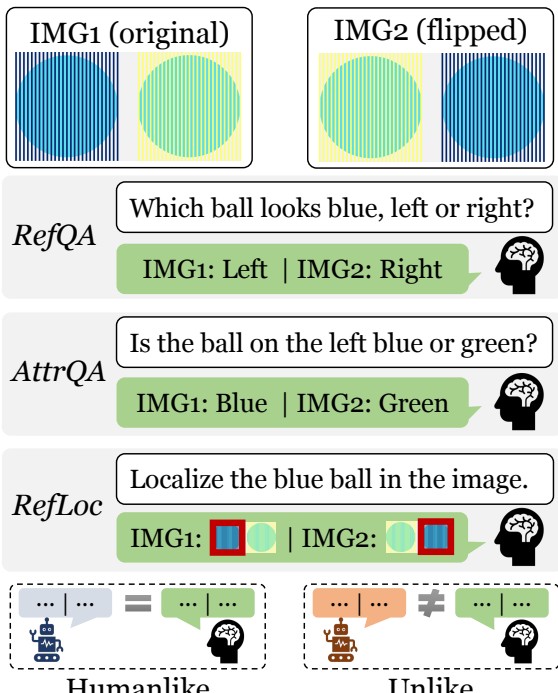

Figure 4: Illustration of the *RefQA*, *AttrQA* and *RefLoc* setups. We flip the illusion image wherein the grounding outcome should also be inverted, to create a pair of images for each test. Model success requires accurate identification in both original and flipped versions to align with human responses. Matching human answers signals the model's capability to interpret illusions in a humanlike way, while a mismatch indicates otherwise.

corresponding annotations are verified by at least three human annotators from our team.

## 4 Experimental Setup

**Vision-Language Models.** To be evaluated on all of the four tasks in GVIL, the model has to be equipped with the visual question-answering skill and the object localization skill simultaneously. Among a few candidates, we choose two state-of-the-art models, the Unified-IO (Lu et al., 2022) and OFA (Wang et al., 2022), both of which are trained on a wide range of vision-language tasks, and achieve impressive performance with a strong zero-shot inference capability on unseen data. Additionally, we select two recent works that adapt large language models to understand visual images: the LLaVA (Liu et al., 2023) and InstructBLIP (Dai et al., 2023). These models are interesting to inspect as they have shown a highly conceptual understanding of the nuance in images, such as the capability of interpreting jokes, which may also be useful in interpreting illusions. For each of the afore-

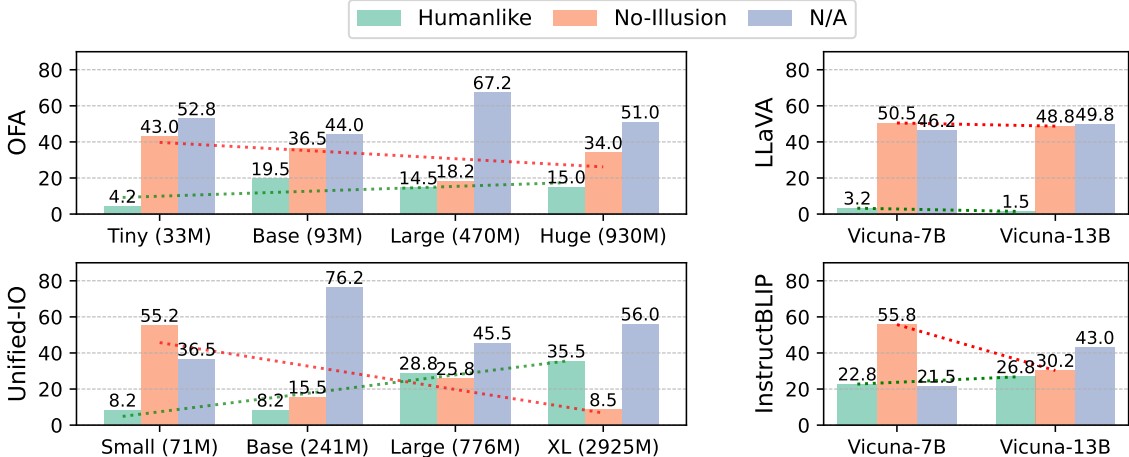

Figure 5: Results of *SameDiffQA*. The number shows the percentage of the answers. Each cluster represents the distribution over humanlike, no-illusion and N/A answers from a model. The green and red line correspond to the linear regression of humanlike rate and no-illusion rate across all the model sizes. Except for OFA-Large, Unified-IO-Large, InstructBLIP-13B, the differences between the humanlike rate and the no-illusion rate are statistically significant $P < 0.005$. Details are in Table 4 Appendix A.

mentioned models, there exists a range of variants in different sizes: OFA-{Tiny, Base, Large, Huge}, Unified-IO-{Small, Base, Large, XL}, LLaVA-{Vicuna-7B, Vicuna-13B}, InstructBLIP-{Vicuna-7B, Vicuna-13B}. This allows us to study the impact of size variations on model's understanding of visual illusions.

**Metrics.** Through the experiments, we keep track of the **Humanlike** rate to measure the alignment between humans and VLMs, which is the percentage of examples where the machine gives exactly the same answers as humans. For the *SameDiffQA* task, we also compute the **No-Illusion** rate, which corresponds to the percentage of examples where the machine consistently considers the objects as the same under both illusion and illusion-free settings. For examples where the model fails to identify the objects as the same in the illusion-free image or produces nonsense answers to the questions, we mark them as **Not Applicable (N/A)** and exclude them from the illusion recognition assessment.

## 5 Results Analysis

From our experiments, we are interested in investigating the following research questions:

- RQ1: to what extent do VLMs recognize the presence of illusions similar to humans?

- RQ2: how much do VLMs align with humans when communication happens under the influence of illusions?

- RQ3: does the degree of alignment between VLMs and human responses vary across different categories of illusions?

We highlight several of our findings across this three questions in below.

### 5.1 Illusion Recognition

The results of *SameDiffQA* are shown in Figure 5. Relative proportions of "humanlike," "no-illusion," and "not applicable (N/A)" responses are represented as green, orange, and grey bars respectively for each model, which all together account for 100%. First of all, we notice a large percentage of responses, across all models, fall under the N/A category. This suggests that these models often cannot even tell that the objects are identical in the illusion-free image, underscoring the need for improvement in standard vision-language reasoning capabilities beyond the scope of illusion contexts.

Given the high proportion of N/A responses, one might question the benchmark's adequacy in reliably reflecting a model's tendency towards either "humanlike" or "no-illusion". Excluding the N/A responses, we employed a $\chi^2$-test and found that 9 out of 12 models would reject the null hypothesis which posits that the "humanlike" or "no-illusion" responses are uniformly distributed. In other words, these models do not behave randomly. Refer to Appendix A for more details. Such findings indicate that, despite certain limitations in their capabilities, our dataset and experimental design effectively

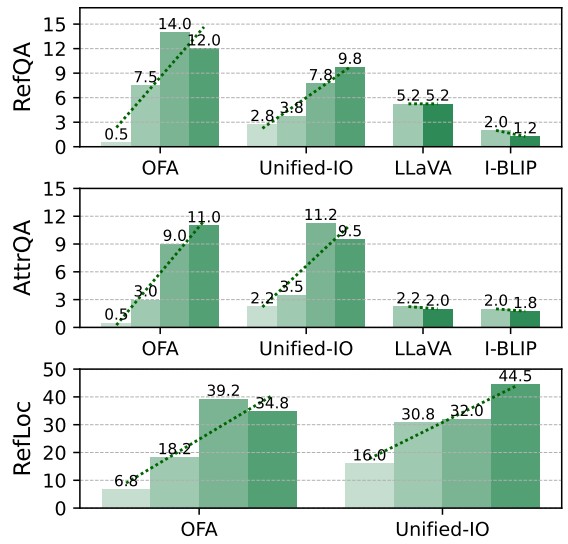

Figure 6: Humanlike rate on *RefQA*, *AttrQA* and *RefLoc*. Each bar represents a different model size, arranged in ascending order from left to right. Note that LLaVA and InstructBLIP cannot predict object bounding boxes thus do not have the *RefLoc* results.

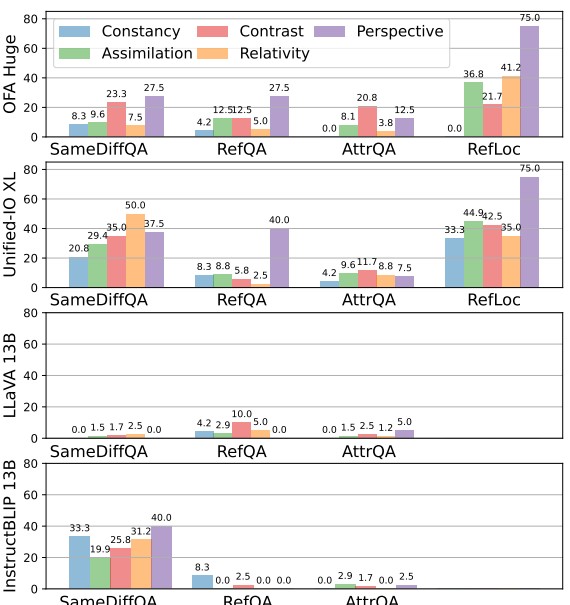

Figure 7: Humanlike rates of the largest model of each family, with finer-grained human-likeness scores on each illusion category.

| Task | Model | Pearson coeff. | p-value |
|------|-------|----------------|---------|
| *SameDiffQA* | OFA | 0.689 | 0.311 |
|  | UnifiedIO | 0.940 | 0.059* |
| *RefQA* | OFA | 0.946 | 0.054* |
|  | UnifiedIO | 0.977 | 0.022** |
| *AttrQA* | OFA | 0.957 | 0.043** |
|  | UnifiedIO | 0.853 | 0.146 |
| *RefLoc* | OFA | 0.933 | 0.066* |
|  | UnifiedIO | 0.960 | 0.039** |

Table 3: Pearson's correlation analysis between the humanlike rate and model size. Statistically significant results with $p < 0.05$ and $p < 0.1$ are marked with ** and *, respectively.

gauge illusion recognition in the assessed VLMs.

When examining cases where responses are applicable for testing illusion recognition, we observe that the majority of models are more likely to fail in recognizing illusions (35.4% on average) than producing humanlike responses (15.6% on average). This discrepancy is most pronounced in the case of InstructBLIP, where the model predominantly offers 'no-illusion' answers. Conversely, the Unified-IO XL model stands out as the only model exhibiting a significant propensity towards humanlike illusion recognition. A further investigation of the underlying reason that causes this discrepancy would be interesting further work.

To illustrate how the scores evolve with model size, we plot regression lines of "humanlike" (green) and "no-illusion" (red) rates, respectively.

An emerging trend reveals that "humanlike" scores tend to increase as the model scales, whereas "no-illusion" responses tend to decline. This finding suggests a positive correlation between model scale and human-machine alignment under illusions. We hypothesize that this observed trend could be attributed to the enhanced pattern-recognition capabilities of larger models. These models, arguably, are better suited to discern and assimilate patterns present in data generated by humans, which may have been shaped by the influence of illusions. Consequently, it's plausible to assume that these models are progressing towards a more humanlike comprehension of illusions.

## 5.2 Communication Under Illusion

The results of *RefQA*, *AttrQA*, and *RefLoc* experiments are shown in Figure 6, offering insights into the alignment between machine and human responses under the influence of visual illusions. We find that all the VLMs encounter significant challenges in responding to questions presented under the influence of visual illusions in both *RefQA* and *AttrQA*. As a result, the models obtain a maximum humanlike response rate of only 14.0% and 11.2% for *RefQA* and *AttrQA*, respectively. Interestingly, models exhibit much stronger alignment in the localization task, with the highest alignment of 44.5% achieved by Unified-IO XL. This indicates that the learned object localization skill aligns

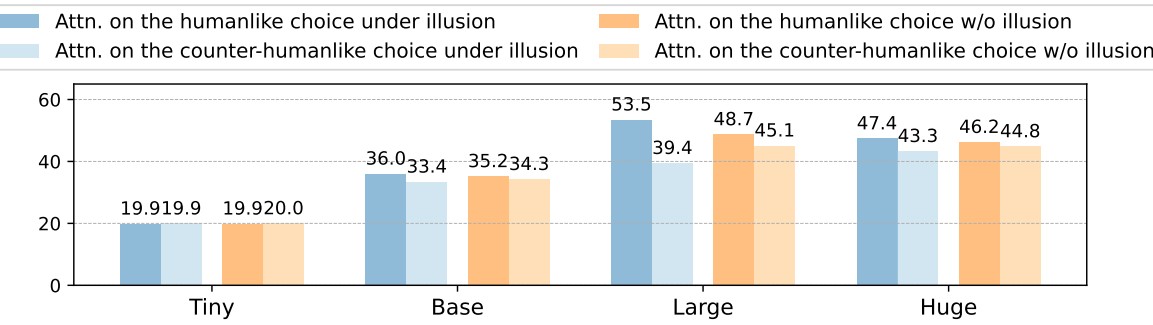

Figure 8: Attention weight distribution of OFA models for the *RefLoc* task.

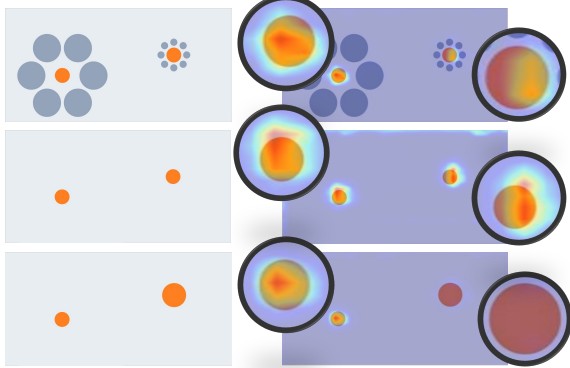

Figure 9: Visualization of the attention maps generated by the OFA-Large model for the *RefLoc* task. In each row, the input image is shown on the left, and the attention map for the referential expression "smaller orange ball" is shown on the right. The attention maps surrounding the object of interest are highlighted for enhanced visibility.

better with humans under illusions compared to the visual question answering skills. Research into the underlying reason behind this difference might be an interesting future direction.

Notably, we find a positive correlation between scaling up models and the increase of humanlike rate across different models and tasks, which echoes our earlier observations from the *SameDiffQA* experiment. To verify the statistical significance, we conducted Pearson's correlation analysis for OFA and UnifiedIO models[2]. As shown in Table 3, 6 of the 8 tested scenarios showed significant or moderately significant positive correlations, with Pearson coefficients exceeding 0.9. Such results underscore the potential of larger models to enhance the human-machine alignment of responses across different tasks and illusion contexts.

[2]InstructBLIP and LLaVA were excluded since at least three data points are needed for the test.

## 5.3 Delving into Illusion Categories

We provide a more granular analysis by examining each type of illusion, presenting the humanlike rates for each category in Figure 7. The results depicted here are sourced from the largest model within each model family, namely Unified-IO Huge, OFA Huge, LLaVA Vicuna-13B, and InstructBLIP Vicuna-13B. Our observation reveals that the *perspective* category demonstrates the highest degree of alignment between machines and humans. On the contrary, color constancy illusions emerge as the category with the least congruity in relation to human responses.

## 5.4 Understanding the Cause of Illusions

To gain insight into model predictions under the influence of illusions, we analyze the attention distributions of OFA models in the *RefLoc* task. Specifically, we compute the attention weight from the localization query (e.g., "the smaller orange ball") to the object representation of either a "humanlike" or "counter-humanlike" perception under illusions. As depicted by the dark blue and light blue bars in Figure 8, as the model size increases, attention weights lean more towards the humanlike selection. This trend is consistent with the humanlike rate observed for the *RefLoc* task in Figure 6. To determine if this bias stems from the illusion, we also calculate attention weights for images without the illusion inducer (represented by orange bars). These weights are nearly equally distributed across both objects, suggesting that the illusion does indeed influence the model's attention and biases its predictions similarly to human perceptions.

Figure 9 shows an example using the attention visualization tool (Aflalo et al., 2022). The first image displays the original illusion image, with two orange balls of identical size while the left ball seems smaller. The second image is devoid

of the illusion inducer, while the third image artificially enlarges the right orange ball. Attention maps corresponding to the "smaller orange ball" query[3] are shown adjacent to each image. In the original illusion, the model predominantly focuses on the left ball, aligning with human observations. Without the illusion inducer, the query becomes ambiguous, leading to a dispersed attention map. However, when an actual size difference is present, the model's attention decisively shifts to the correctly perceived smaller ball on the left. A comparison of these attention maps highlights that while illusions can steer the model's attention similarly to humans, its effect is less pronounced than when a real disparity exists.

## 6  Discussion and Conclusion

We introduce GVIL, the first dataset facilitating a systematic evaluation of machine visual illusion via natural language. Evaluating four distinct series of state-of-the-art vision-language model families across varying scales, we observe a notable alignment between these models and human perceptions during object localization in the context of illusions. Interestingly, this alignment tends to strengthen with increased model size. Conversely, many models face challenges in mirroring human perspectives within visual question-answering tasks. Our preliminary observations underscore the need for further discussions in two directions:

**Assessment of Vision-Language Models in the Realm of Visual Illusions.** Vision-language models have demonstrated commendable prowess in both visual and language understanding. Yet, a notable gap persists in assessing their performance in the presence of visual illusions. Given that such illusions are intrinsic to human perception, overlooking this facet may contribute to misalignment between human and AI interpretations during real-world engagements. While our study unveils certain trends, like the correlation between model size and human-model alignment, making definitive assertions is non-trivial due to the discrepancy in model architectures and their training datasets. Through GVIL, we aspire to catalyze further research that addresses visual illusion in VLMs.

---

[3]We use the second last layer of the OFA large model, as the overall attention score of this layer is the highest. Attentions from all the heads are averaged.

**Gaining Insights from Illusions.** Exploring the effects of visual illusions can offer fresh perspectives to comprehend the intricacies of vision-language models. Visual illusion, in some way, is similar to various types of values shared by our human society, but not shared among today's AI models. Given the rapidly growing applications of large AI models, it's important to identify and understand various aspects of alignment between these models and humans. Vision illusion is only an example among many possibilities for future studies.

## Limitations

This work is only the initial attempt to the question and there are many limitations which we think of as exciting future research directions. First of all, although our experiments yields some interesting empirical findings, it is not clear why different forms of tasks (e.g., QA-based tasks vs. RefLoc) lead to a stark contrast in the results. As these findings may have implications in future technology that adapt to visual illusions, more in-depth understanding of these behaviors will be needed in the future. Second, our benchmark is currently small in size. It lays an infrastructure for this work. Future efforts to collect more data to form a centralized repository will be desired for studying visual illusions in both humans and machines. Third, our investigation is only based on a manually collected dataset for our intellectual curiosity. The construction of this dataset has the limitations that the effect of visual illusions are not validated by a wider range of human subjects other than the authors. While it has motivation in improving situated language communication with embodied agents in the physical world, how visual illusions play in perceiving and communicating about the real physical world remains an interesting question.

## Ethics Statement

The data are collected and annotated by the authors without the involvement of any other human subject. Data samples are selected from a wide literature search on the subject of visual illusions.

## Acknowledgements

This work was supported by NSF IIS-1949634 and the DARPA PTG program HR00112220003. The authors would like to thank the anonymous reviewers for their valuable comments and suggestions.

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

# A  Statistical Analysis for Illusion Recognition

We conducted a statistical analysis to rigorously validate that our experimental setup is able to reveal a model's inclination towards either "humanlike" or "no-illusion", notwithstanding the high prevalence of N/A samples. Specifically, we applied a $\chi^2$-test to the model predictions omitting the N/A samples. The null hypothesis posits that the humanlike and no-illusion samples are uniformly distributed, i.e., the model behaviors randomly. As shown in Table 4, a significant proportion of the models reject the null hypothesis. Out of the 12 models tested, 8 models rejected the null hypothesis with a $p$-value $< 0.001$, and 9 models rejected the null hypothesis with a $p$-value $< 0.01$. These figures strongly suggest that most models perform better than what would be expected by chance alone, which is a piece of strong evidence that our dataset and experimental setup can support the evaluation of illusion recognition for the tested VLMs.

| Model | #HL | #NI | $\chi^2$ | p-value |
|---|---|---|---|---|
| OFA-Tiny | 17 | 172 | 129.45 | <0.001*** |
| OFA-Base | 78 | 146 | 20.64 | <0.001*** |
| OFA-Large | 58 | 73 | 1.72 | 0.190 |
| OFA-Huge | 60 | 136 | 29.47 | <0.001*** |
| UnifiedIO-Small | 33 | 221 | 139.15 | <0.001*** |
| UnifiedIO-Base | 33 | 62 | 9.38 | 0.002** |
| UnifiedIO-Large | 115 | 103 | 0.66 | 0.416 |
| UnifiedIO-XL | 142 | 34 | 66.27 | <0.001*** |
| LLaVA-7B | 13 | 202 | 12.89 | <0.001*** |
| LLaVA-13B | 6 | 195 | 177.72 | <0.001*** |
| InstructBLIP-7B | 91 | 223 | 55.49 | <0.001*** |
| InstructBLIP-13B | 107 | 121 | 0.86 | 0.354 |

Table 4: Chi-square test for the SameDiff Task (Figure 5). #HL and #NI denote the number of humanlike illusionary answers and no-illusion answers, respectively. Statistically significant results with p < 0.001 and p < 0.05 are marked with *** and **, respectively.