# OpenReview forum: "Grounding Visual Illusions in Language: Do Vision-Language Models Perceive Illusions Like Humans?"
_EMNLP/2023/Conference — EMNLP 2023 Main_

### Official Review · Reviewer_Zh5s · 2023-07-21

**Typos Grammar Style And Presentation Improvements:** Very well written.
**Soundness:** 5

**Excitement:**

4: Strong: This paper deepens the understanding of some phenomenon or lowers the barriers to an existing research direction.

**Paper Topic And Main Contributions:**

This paper tests whether VLMs are also prone to visual illusions like humans are. It finds that indeed, they are to a small degree.
For this, the authors assemble a new VL dataset testing for visual illusions from 4 categories in a QA fashion and 4 recent test VL models on it.

**Questions For The Authors:**

Will you release the code and dataset?

It is not really a question, more of a comment, so no need to address this in the rebuttal if space is limited:

VLMs are trained on images and text where the text has been generated by humans alongside the image? Isn’t it expected for the models to show a certain degree of visual illusions because they have been learning to describe images like humans and give answers human expect? I am surprised to see that they produce little humanlike responses. Maybe this is correlated with their poor performance in general, that most of the time (L410) they do not even recognize the object, so recognizing the “ambiguitiy” of a visual illusion might require even more capabilities that the models do not have? This thought seems to align with what you write in L472.

**Reasons To Accept:**

The contributions of this paper are substantial:

1.	Introduction of a visual illusion dataset for VLMs including visual illusion from 4 categories.

2.	Testing 4 SOTA VLMs.

3.	Clearly written, visualised and well motivated paper.

The research question is interesting and the experiments are well executed to address it.

**Reasons To Reject:**

This paper makes it really hard to find a reason to reject, as its contribution is substantial and the methodology is sound.

**Reproducibility:**

4: Could mostly reproduce the results, but there may be some variation because of sample variance or minor variations in their interpretation of the protocol or method.

**Reviewer Confidence:**

4: Quite sure. I tried to check the important points carefully. It's unlikely, though conceivable, that I missed something that should affect my ratings.

---

> ### Author Rebuttal · Authors · 2023-08-29
>
> We thank the reviewer for acknowledging the contribution of this paper is “substantial”. We appreciate the reviewer’s valuable feedback and questions which call for more in-depth studies in the future.
>
> > **Will you release the code and dataset?**
> - Absolutely. We will release the dataset and software after the anonymity period.
>
> > **VLMs are trained on images and text where the text has been generated by humans alongside the image? Isn’t it expected for the models to show a certain degree of visual illusions because they have been learning to describe images like humans and give answers human expect? I am surprised to see that they produce little humanlike responses. Maybe this is correlated with their poor performance in general, that most of the time (L410) they do not even recognize the object, so recognizing the “ambiguity” of a visual illusion might require even more capabilities that the models do not have? This thought seems to align with what you write in L472.**
> - We appreciate the reviewer for pointing this out. One of the driving-force of conducting this study is our curiosity about whether a VLM that is competitive on common vision-language benchmarks is better at aligning with humans even when they are deceived, or better at faithfully representing the actual reality. From the trend of an increased human likeness when the model becomes larger, it seems that the model does learn to mimic more human behaviors when fitting the data produced by humans. Regarding the low ratio of human-like answers, just as you mentioned, it could be that the models are too poor to output robust and self-consistent answers, in that sense not humanlike. Another possibility is that the models may not be trained on many illusion images, thus struggling in those out-of-distribution samples. These are interesting questions which will require more in-depth studies in the future.

---

### Official Review · Reviewer_EWKP · 2023-08-02

**Paper Topic And Main Contributions:** 1. Constructed a visual illusion data…
**Soundness:** 4

**Excitement:**

4: Strong: This paper deepens the understanding of some phenomenon or lowers the barriers to an existing research direction.

**Reasons To Accept:**

1. The problem about visual illusion of VLM is interesting.

2. The paper makes a focused contribution.

**Reasons To Reject:**

Significant amount of the VLMs’ output is N/A which indicates the incapability of VLMs on more foundation tasks and make the result and analysis less convincing.

**Reproducibility:**

5: Could easily reproduce the results.

**Reviewer Confidence:**

3: Pretty sure, but there's a chance I missed something. Although I have a good feel for this area in general, I did not carefully check the paper's details, e.g., the math, experimental design, or novelty.

---

> ### Author Rebuttal · Authors · 2023-08-29
>
> We thank the reviewer for recognizing that visual illusions of VLM is an interesting problem and this paper makes a focused contribution. We appreciate the reviewer’s feedback on the evaluation results. We believe our response has sufficiently addressed the reviewer’s concern. We will be more than happy to answer more questions if any part of the response is not clear.
>
> > **Significant amount of the VLMs’ output is N/A which indicates the incapability of VLMs on more foundation tasks and make the result and analysis less convincing.**
> - First, we acknowledge that the current VLMs are far from being perfect on some foundation tasks as indicated by the high N/A ratio. However, the goal of our work is to (1) define the problem of machine visual illusion in the context of VLM; and (2) create a first-of-its-kind dataset that can support future work on visual illusion. Although the SOTA models we applied are not reliable or robust enough, the problem itself and the dataset are not specific to these models. As the field progresses and new models are introduced at an unprecedented rate, this work is to set the foundation/stage where future models can be evaluated upon.
> - Second, even with the existing models, this work has generated some key findings. The high ratio of N/A samples in the SameDiffQA task does not affect the validity of visual recognition evaluation.  Here we show statistical analysis for the results in Figure 5. We did a Chi-square test among the samples that excluded the “N/A” samples. The null hypothesis is that the humanlike or no-illusion samples are uniformly distributed, i.e., the model behaves randomly. As shown in Table, 8 out of 12 models reject the null hypothesis with a p-value <0.001, and  9 out of 12 reject with a p-value <0.01. This suggests that most models do not merely perform at chance level. This indicates strong evidence that our dataset and experimental setup can support the evaluation of illusion recognition for the tested VLMs even with limited capability of answering these questions. We will add these analysis results in the revised version.
>
> |        Model      | #Humanlike | #No-illusion |Chi-square|  p-value|
> |-----------------------|--------------|-----------------|---------------|------------|
> | OFA-tiny             |      17      |        172      |    129.45   |   <0.001 |
> | OFA-base           |      78      |        146      |    20.64     |  <0.001 |
> | OFA-large           |      58      |        73        |    1.72       |  0.190    |
> | OFA-huge           |      60      |        136      |    29.47     |   <0.001 |
> | UnifiedIO-small   |      33      |        221      |    139.15   |   <0.001 |
> | UnifiedIO-base   |      33       |        62        |    9.38       |    0.002  |
> | UnifiedIO-large   |      115     |       103       |    0.66       |  0.416    |
> | UnifiedIO-xl         |     142     |        34        |     66.27     |  <0.001  |
> | LLaVA-7B            |     13       |        202      |    12.89      |  <0.001  |
> | LLaVA-13B          |     6         |        195      |    177.72    |  <0.001  |
> | InstructBLIP-7B   |     91       |        223      |    55.49      |   <0.001 |
> | InstructBLIP-13B |     107     |        121      |    0.86        |    0.354  |
>
> Table 1. Chi-square test for the SameDiff Task (Figure 5)

---

### Official Review · Reviewer_2Xp9 · 2023-08-09

**Typos Grammar Style And Presentation Improvements:** Figure 6
**Soundness:** 4

**Excitement:**

4: Strong: This paper deepens the understanding of some phenomenon or lowers the barriers to an existing research direction.

**Paper Topic And Main Contributions:**

This paper examines the performance of Vision-and-Language models (VLMs) in addressing questions involving visual illusions. A new dataset, GVIL, comprising five types of visual illusions, is assembled. Multiple VLMs are tested on four tasks to showcase their behavior. The results reveal that these models closely match human performance in object localization, yet face challenges in aligning with humans for visual question-answering scenarios.

**Questions For The Authors:**

see weakness

**Reasons To Accept:**

- Exploring the behavior of VLMs in the context of machine-generated visual illusions is intriguing and holds promise for inspiring future research.
- Overall, the paper is well-written. The comprehensive experiments and analysis can illustrate some behaviors of prominent VLMs.

**Reasons To Reject:**

- The size of the proposed dataset is limited. This raises concerns about the reliability of the evaluation findings.
- Statistical tests were not carried out. Fig. 5 illustrates that the N/A (not applicable) sample rate is approximately 50%, suggesting that the evaluated VLM struggles to comprehend and answer questions accurately. This leaves room for chance-level responses, human-like or not. While the authors introduce a paired test (flipping the sample), I find it insufficient. I recommend the authors evaluate with more augmented samples and perform statistical tests to further validate their conclusion.
- While this paper sheds light on the behavior of certain VLMs, it lacks an analysis of the underlying reasons for these behaviors.

**Reproducibility:**

4: Could mostly reproduce the results, but there may be some variation because of sample variance or minor variations in their interpretation of the protocol or method.

**Reviewer Confidence:**

3: Pretty sure, but there's a chance I missed something. Although I have a good feel for this area in general, I did not carefully check the paper's details, e.g., the math, experimental design, or novelty.

---

> ### Author Rebuttal · Authors · 2023-08-29
>
> We thank the reviewer for recognizing this work is “intriguing” and “holds promise for inspiring future work”. We appreciate all valuable feedback and questions about evaluation results. We believe we have addressed the reviewer’s concerns by providing detailed statistical analysis results. We welcome more questions during the discussion period if any part of our response is not clear.
>
> > **The size of the proposed dataset is limited. This raises concerns about the reliability of the evaluation findings.**
> - We acknowledge the current size of the data is on the small side, however it’s sufficient to support several initial findings (see the results next).  We want to emphasize that GVIL, while limited in its size, is the first-of-its-kind dataset that was carefully curated to facilitate study of visual illusions by VLMs. The diverse set of illusions included in the dataset ensures that it's not just the quantity but also the quality and complexity of data that provides value, especially as a first step to initiating this research direction.  Nevertheless, we aim to expand the dataset in our future work and make a platform available for the community to contribute to data expansion.
>
> > **Concerns about whether the high ratio of N/A samples in the SameDiffQA task (Fig. 5) will affect the conclusion's validity.**
> - We perform a statistical analysis to validate that our experimental setup is able to show the model’s behavior towards either humanlike or no-illusion, even with a high N/A ratio. In particular, we do a Chi-square test among the samples that exclude the “N/A” samples. The null hypothesis is that the humanlike or no-illusion samples are uniformly distributed, i.e., the model behaviors randomly. As shown in Table, 8 out of 12 models reject the null hypothesis with a p-value <0.001, while 9 out of 12 reject with a p-value <0.01. This suggests that most models do not merely perform at chance level. This indicates strong evidence that our dataset and experimental setup can support the evaluation of illusion recognition for the tested VLMs even with limited capability of answering these questions. We will add this statistical analysis in the revision.
>
> |        Model      | #Humanlike | #No-illusion |Chi-square|  p-value|
> |-----------------------|--------------|-----------------|---------------|------------|
> | OFA-tiny             |      17      |        172      |    129.45   |   <0.001 |
> | OFA-base           |      78      |        146      |    20.64     |  <0.001 |
> | OFA-large           |      58      |        73        |    1.72       |  0.190    |
> | OFA-huge           |      60      |        136      |    29.47     |   <0.001 |
> | UnifiedIO-small   |      33      |        221      |    139.15   |   <0.001 |
> | UnifiedIO-base   |      33       |        62        |    9.38       |    0.002  |
> | UnifiedIO-large   |      115     |       103       |    0.66       |  0.416    |
> | UnifiedIO-xl         |     142     |        34        |     66.27     |  <0.001  |
> | LLaVA-7B            |     13       |        202      |    12.89      |  <0.001  |
> | LLaVA-13B          |     6         |        195      |    177.72    |  <0.001  |
> | InstructBLIP-7B   |     91       |        223      |    55.49      |   <0.001 |
> | InstructBLIP-13B |     107     |        121      |    0.86        |    0.354  |
>
> Table 1. Chi-square test for the SameDiff Task (Figure 5).
>
> &nbsp;
>
> **Additional statistical test for the correlation between human likeness and model sizes.**
> - To validate our second main claim that the model’s human likeness under visual illusions increases with their size, we also add a Pearson’s correlation analysis on all the four tasks for OFA and UnifiedIO. Note that we need at least 3 data points to perform the test so InstructBLIP and LLaVA are not applicable here. As shown in Table 2, under 6 out of 8 setups we observe a significant or moderate significant result for a positive correlation (pearson coefficient >0.9). This supports our finding of positive correlation between scaling up models and the increase of humanlike rate across different models and tasks.
>
> |        Task             |   Model    |  Pearson coefficient   |  p-value |
> |-----------------------|--------------|------------------------------|-------------|
> | SameDiff QA      |      OFA    |             0.689              |  0.311    |
> | SameDiff QA      | UnifiedIO |             0.940              |  0.059*  |
> |          Ref QA      |      OFA    |             0.946              |  0.054*  |
> |          Ref QA      | UnifiedIO |             0.977              |  0.022** |
> |          Attr QA      |      OFA    |             0.957              |  0.043** |
> |          Attr QA      | UnifiedIO |             0.853              |  0.146    |
> |          Ref Loc     |      OFA    |             0.933              |  0.066*   |
> |          Ref Loc     | UnifiedIO |             0.960              |  0.039**  |
>
> Table 2. Pearson's correlation analysis between the humanlike rate and model size.   **: p < 0.05   *: p<0.1
>
> &nbsp;
>
> > **Lacks an analysis of the underlying reasoning of machine illusions**
> - We fully acknowledge the reviewer's point regarding the importance of a deeper analysis of the underlying behaviors of the VLMs. In Section 5.4 and Figure 8, we attempted to address this question by showing the visualized attention map of an OFA model under different situations. We observe that the model’s attention weight can be biased towards a humanlike manner with the presence of illusion inducers, which explains its output to some degree. While understanding the precise 'why' behind these behaviors is crucial, we intended to set the stage for this exploration. We hope the preliminary results shown in our paper and the resources that will be released soon can inspire future research to understand the problem better.
>
> > **Figure 6: "14.0" is out of the figure**
> - Thanks for pointing out this. We will correct the figure in the updated version.

---

### Meta-Review · Area_Chair_bByf · 2023-09-18

**Recommendation:** 5

**Metareview:**

Reviewers judged this work to be sound technically and also reproducible. The paper is well written and clear. The study itself was found to be creative and novel, with important results. The major limitation is the small size of the dataset and the fact that the VLM used this work would respond with "N/A" is a large number of trials.

---

### Decision · Program_Chairs · 2023-10-07

**Decision:**

Accept-Main

**Comment:**

Reviewers judged this work to be sound technically and also reproducible. The paper is well written and clear. The study itself was found to be creative and novel, with important results. The major limitation is the small size of the dataset and the fact that the VLM used this work would respond with "N/A" is a large number of trials.